# Ecology and evolution of viruses infecting uncultivated SUP05 bacteria as revealed by single-cell- and meta-genomics

Simon Roux[1], Alyse K Hawley[2], Monica Torres Beltran[2], Melanie Scofield[2], Patrick Schwientek[3], Ramunas Stepanauskas[4], Tanja Woyke[3], Steven J Hallam[2,5]*, Matthew B Sullivan[1]*

[1]Department of Ecology and Evolutionary Biology, University of Arizona, Tucson, United States; [2]Department of Microbiology and Immunology, University of British Columbia, Vancouver, Canada; [3]U.S Department of Energy Joint Genome Institute, Walnut Creek, United States; [4]Bigelow Laboratory for Ocean Sciences, East Boothbay, United States; [5]Graduate Program in Bioinformatics, University of British Columbia, Vancouver, Canada

**Abstract** Viruses modulate microbial communities and alter ecosystem functions. However, due to cultivation bottlenecks, specific virus–host interaction dynamics remain cryptic. In this study, we examined 127 single-cell amplified genomes (SAGs) from uncultivated SUP05 bacteria isolated from a model marine oxygen minimum zone (OMZ) to identify 69 viral contigs representing five new genera within dsDNA *Caudovirales* and ssDNA *Microviridae*. Infection frequencies suggest that ~1/3 of SUP05 bacteria is viral-infected, with higher infection frequency where oxygen-deficiency was most severe. Observed *Microviridae* clonality suggests recovery of bloom-terminating viruses, while systematic co-infection between dsDNA and ssDNA viruses posits previously unrecognized cooperation modes. Analyses of 186 microbial and viral metagenomes revealed that SUP05 viruses persisted for years, but remained endemic to the OMZ. Finally, identification of virus-encoded dissimilatory sulfite reductase suggests SUP05 viruses reprogram their host's energy metabolism. Together, these results demonstrate closely coupled SUP05 virus–host co-evolutionary dynamics with the potential to modulate biogeochemical cycling in climate-critical and expanding OMZs.

*For correspondence: shallam@mail.ubc.ca (SJH); mbsulli@email.arizona.edu (MBS)

**Competing interests:** The authors declare that no competing interests exist.

**Reviewing editor**: Nicole Dubilier, Max Planck Institute for Marine Microbiology, Germany

## Introduction

Microbial communities are critical drivers of nutrient and energy conversion process in natural and engineered ecosystems (*Falkowski et al., 2008*). In the last two decades, it has progressively become clear that viral-mediated predation, gene transfer, and metabolic reprogramming modulate the structure, function, and evolutionary trajectory of these microbial communities (*Suttle, 2007*; *Abedon, 2009*; *Rodriguez-Valera et al., 2009*; *Hurwitz et al., 2013*). At the same time, the vast majority of microbes and viruses remain uncultivated and their diversity is extensive, so that model system-based measurements rarely reflect the network properties of natural microbial communities. While culture-independent methods, such as metagenomics and metatranscriptomics, can illuminate latent and expressed metabolic potential of microbial (*Frias-Lopez et al., 2008*; *Venter et al., 2004*; *Stewart et al., 2012*; *DeLong et al., 2006*) or viral communities (*Angly et al., 2006*; *Hurwitz et al., 2013*; *Mizuno et al., 2013*), interactions between community members remain difficult to resolve.

Clustered regularly interspaced short palindromic repeats (CRISPRs) containing short stretches of viral or plasmid DNA separated between repeat sequences can provide a record of past infections in uncultivated microbial communities. Together with associated Cas (CRISPR-associated) genes, CRISPRs

**eLife digest** Microorganisms help to drive a number of processes that recycle energy and nutrients, including elements such as carbon, nitrogen, and sulfur, around the Earth's ecosystems. Viruses that infect microbes can also affect these cycles by killing and breaking open microbial cells, or by reprogramming the cell's metabolism. However, as there are many different species of microbes and viruses —the vast majority of which cannot easily be grown in the laboratory— little is known about most virus–host interactions in natural ecosystems, especially in the oceans.

In the world's oceans, the concentration of oxygen dissolved in the water changes in different regions and at different depths. 'Oxygen minimum zones' occur globally throughout the oceans at depths of 200–1000 meters, and climate change is causing these zones to expand and intensify. Although a lack of oxygen is sometimes considered detrimental to living organisms, oxygen minimum zones appear to be rich with microbial life that is adapted to thrive under oxygen-starved conditions.

Sulfur-oxidizing bacteria are one of the most abundant groups of microbes in these oxygen minimum zones, and several of these bacteria are known to influence the recycling of chemical substances. Now, Roux et al. introduce a new method to identify viruses that infect the microbes in this environment, including those microbes that cannot be grown in the laboratory and which have previously remained largely unexplored.

The genomes of 127 individual bacterial cells —collected from an oxygen minimum zone in western Canada— were examined. Roux et al. estimate that about a third of the sulfur-oxidizing bacterial cells are infected by at least one virus, but often multiple viruses infected the same bacterium. Five new genera (groups of one or more species) of viruses were also discovered and found to infect these bacteria. Looking for these new viral sequences in the DNA of this oxygen minimum zone's microbial community revealed that these newly discovered viruses persist in this region over several years. It also revealed that these viruses appear to only be found within the oxygen minimum zone. Roux et al. uncovered that these viruses carry genes that could manipulate how an infected bacterium processes sulfur-containing compounds; this is similar to previous observations showing that other viruses also influence cellular process (such as photosynthesis) in infected bacteria. As such, these newly discovered viruses might also influence the recycling of chemical elements within oxygen minimum zones.

Together, Roux et al.'s findings provide an unprecedented look into a wild virus community using a method that can be generalized to uncover viruses in a data type that is quickly becoming more widespread: single cell genomes. This effort to understand virus–host interactions by looking in the genomes of individual cells now sets the stage for future efforts aimed to uncover the impact of viruses on bacteria in other environments across the globe.

function as an adaptive immune system in prokaryotes with the potential to suppress viral replication or horizontal gene transfer (*Sorek et al., 2008*). However, an application of CRISPR-based virus–host association to both uncultivated hosts and viruses require the assembly of complete or near-complete genomes of both entities, limiting their utility to lower diversity ecosystems (*Andersson and Banfield, 2008*; *Anderson et al., 2011*). Alternatively, single-cell amplified genome (SAG) sequencing is emerging as a more direct method to chart metabolic potential of individual cells within microbial communities with special emphasis on candidate phyla that have no cultured representatives (*Yoon et al., 2011*; *Martinez-Garcia et al., 2012*; *Rinke et al., 2013*; *Swan et al., 2013*). Here, we combine metagenomic and single-cell genomic sequencing to explore virus–host interactions within uncultivated bacteria inhabiting a marine oxygen minimum zone (OMZ).

Marine OMZs, defined by dissolved oxygen concentrations <20 µmol kg$^{-1}$, are oceanographic features that arise from elevated demand for respiratory oxygen in poorly ventilated, highly stratified waters. OMZs are crucial for biogeochemical cycles in the global ocean, as they represent hotspots for microbial-driven carbon, nitrogen, and sulfur transformations (*Ulloa et al., 2012*; *Wright et al., 2012*) and play a disproportionate role in nitrogen loss processes and greenhouse gas cycling (*Lam et al., 2009*; *Ward et al., 2009*). Moreover, these zones are expanding due to changing ocean water temperatures and circulation patterns (*Stramma et al., 2008*; *Whitney et al., 2007*). Given these changing physical and chemical conditions and the importance of OMZs to ocean-atmosphere functioning,

a clearer understanding of biological responses is critical to develop a much-needed predictive modeling capacity for OMZs.

In OMZs, microbial communities drive matter and energy transformations and are typically dominated by sulfur-oxidizing *Gammaproteobacteria* related to the chemoautotrophic gill symbionts of deep-sea clams and mussels (*Stewart et al., 2012*; *Wright et al., 2012*). Phylogenetic analysis indicates that these bacteria are comprised of two primary lineages; one consisting of sequences affiliated with SUP05 and clam and mussel symbionts, and the other consisting of sequences affiliated with Arctic96BD-19 (*Walsh et al., 2009*; *Wright et al., 2012*). Both groups partition along gradients of oxygen and sulfide, with Arctic96BD-19 most prevalent in oxygenated waters and SUP05 most prevalent in anoxic or anoxic/sulfidic waters (*Wright et al., 2012*). Niche partitioning between SUP05 and Arctic96BD-19 is driven by complementary modes of carbon and energy metabolism that harness alternative terminal electron acceptors. While both Arctic96BD-19 and SUP05 use reduced sulfur compounds as electron donors to drive inorganic carbon fixation, SUP05 manifests a more versatile energy metabolism linking carbon, nitrogen, and sulfur cycling within OMZ and hydrothermal vent waters (*Canfield et al., 2010*; *Zaikova et al., 2010*; *Swan et al., 2011*; *Stewart et al., 2012*; *Anantharaman et al., 2013*; *Mattes et al., 2013*; *Anantharaman et al., 2014*; *Hawley et al., 2014*).

Ocean viruses, predominantly investigated in the sunlit or photic zone, are abundant, dynamic, and diverse (*Suttle, 2005*) with growing evidence for direct roles in metabolic reprogramming of microbial photosynthesis, central carbon metabolism, and sulfur cycling (*Mann et al., 2003*; *Lindell et al., 2005*; *Clokie et al., 2006*; *Breitbart et al., 2007*; *Dammeyer et al., 2008*; *Sharon et al., 2009*, *2011*; *Thompson et al., 2011*; *Hurwitz et al., 2013*). Preliminary studies suggest that similar patterns are emerging in OMZ waters. In the Eastern Tropical South Pacific, a metagenomic survey revealed specific viral populations endemic to OMZ waters (*Cassman et al., 2012*). Consistent with most viral metagenome surveys, approximately 3% of sequences were affiliated with functionally annotated genes in public databases. From a nitrogen and sulfur cycling perspective, viromes from the oxycline contained genes encoding components of nitric oxide synthase, nitrate and nitrite ammonification, and ammonia assimilation pathways as well as inorganic sulfur assimilation (*Cassman et al., 2012*). In anoxic waters, viromes contained genes encoding components of denitrification, nitrate and nitrite ammonification, and ammonia assimilation pathways as well as sulfate reduction, thioredoxin-disulfide reductase, and inorganic sulfur assimilation (*Cassman et al., 2012*). More recently, metagenomic analyses of hydrothermal vent plume microbial communities dominated by SUP05 bacteria-enabled phage genome assemblies presumed to infect SUP05 (*Anantharaman et al., 2014*). Consistent with viruses encoding auxiliary metabolic genes (AMGs, *Breitbart et al., 2007*) enabling viral reprogramming of microbial metabolic pathways (*Lindell et al., 2005*; *Thompson et al., 2011*), putative SUP05 phage contained genes encoding reverse dissimilatory sulfite reductase A and C positing a role for viruses in modulating the marine sulfur cycle (*Anantharaman et al., 2014*).

Given that SUP05 and Arctic96BD-19 play key roles in OMZ ecology and biogeochemistry, we designed an approach to target SUP05-associated viruses in a model OMZ ecosystem, Saanich Inlet a seasonally anoxic fjord on the coast of Vancouver Island, British Columbia, Canada. We obtained a SUP05 single-cell genomic data set spanning defined redox gradients in the Saanich Inlet water column, identified SUP05-associated viruses infecting SAGs, and used resulting virus–host pairs as recruitment platforms to estimate viral diversity, activity, dispersion, and potential impact on SUP05 population dynamics and metabolic capacity. The resulting data sets open an unprecedented window on uncultivated virus–host dynamics in OMZs and provide an analytical approach extensible to other natural or engineered ecosystems.

## Results and discussion

### Generating a SUP05 bacterial genomic data set

SUP05 SAGs were generated at the Bigelow Laboratory for Ocean Sciences (http://scgc.bigelow.org, [*Stepanauskas and Sieracki, 2007*; *Swan et al., 2013*]). Briefly, fluorescence-activated cell sorting was used to separate individual cells <10 µm in diameter from 100, 150, and 185 meters water depth, spanning water column gradients of oxygen and sulfide in Saanich Inlet (*Figure 1—figure supplement 1*). Water column redox conditions were typical for stratified summer months when SUP05 populations bloom in deep basin waters. A total of 315 anonymously sorted cells (discriminated solely using fluorescence and size for sorting) per depth interval were subjected to multiple displacement amplification

(MDA), and the taxonomic identity of single amplified genomes (SAGs) was determined by directly sequencing bacterial small subunit ribosomal RNA (SSU rRNA) gene amplicons. SAGs affiliated with SUP05 (n = 127) and Arctic96BD-19 (n = 9) populations were subsequently whole genome shotgun sequenced on the Illumina HiSeq platform. Most (113/127) SUP05 SAGs fell into two major operational taxonomic units (OTUs) or subclades, based on SSU rRNA gene sequence clustering at the 97% identity threshold—SUP05_01 (n = 65) and SUP05_03 (n = 48) (*Figure 1—figure supplement 2*). SUP05_01 SAGs were recovered at 100, 150, and 185 meters, peaking at 150 meters, while SUP05_03 SAGs were more evenly distributed between 150 and 185 meters. A number of SUP05 SAG assemblies contained viral contigs consistent with sampling infected cells across the redoxcline.

## New SUP05-associated phage genomes

50 *bona fide* viral contigs (*Supplementary file 1*, 'Materials and methods') were identified in 30 SUP05 SAGs using viral marker genes, hereafter termed 'hallmark genes' (*Abrescia et al., 2012*). SUP05 viral contigs were affiliated with known families of *Caudovirales* (dsDNA) and *Microviridae* (ssDNA) bacteriophages. The presence of *Caudovirales* is not surprising as they are commonly observed in oceanic samples (*Williamson et al., 2012*; *Hurwitz and Sullivan, 2013*), including the ETSP OMZ and SUP05-dominated hydrothermal vent plumes (*Cassman et al., 2012*; *Anantharaman et al., 2014*). *Microviridae,* however, are usually observed in surface seawater or deep-sea sediments and have not been previously associated with OMZs (*Angly et al., 2009*; *Tucker et al., 2011*; *Yoshida et al., 2013*; *Labonté and Suttle, 2013b*). Given the SUP05 lineages described above, we note that viral contigs recovered from SUP05_01 SAGs were exclusively *Caudovirales,* whereas SUP05_03 SAGs contained both *Caudovirales* and *Microviridae.* Using non-reference-based methods, an additional 19 contigs were identified as putative viral sequences. These sequences did not encode hallmark genes, but displayed genomic characteristics consistent with novel viral genomes including a low ratio of characterized genes (i.e., most genes predicted on these contigs do not match any sequences from the reference databases), a high number of short genes, and a low number of strand changes between two consecutive genes (i.e., gene sets tend to be coded on the same strand; 'Materials and methods', *Figure 1— figure supplement 3*). In total, 69 viral contigs encoding 898 predicted open reading frames over 529 kb were recovered from SUP05 SAGs representing current viral infections.

## Viral infection of SUP05 cells in nature

Forty-two out of 127 SUP05 SAGs sequenced contained one or more viral contigs (*Figure 1—source data 1*), indicating that ~1/3 of SUP05 cells inhabiting the Saanich Inlet water column were infected by viruses. Such lineage-specific infection frequency determination is unprecedented in uncultivated or cultivated host cells and is largely consistent with community-averaged estimates for marine bacteria (*Suttle, 2007*). As with all the other means to estimate infection frequency and viral-induced microbial mortality (*Brum et al., 2014*), there are caveats to these numbers including underestimation linked to incomplete identification of viruses in the SAG data sets. Such an underestimation could result from (i) lack of reference genomes, (ii) incomplete SAG genomes, (iii) early infections not being detected prior to genome insertion and replication, or (iv) late infections not being detected due to phage-directed degradation of host DNA preventing 16S identification during the SAG selection process. Since the infection frequency estimates are largely consistent with community-based measurements, we expect that these biases are small.

SUP05 viral infections showed strong depth partitioning along defined gradients of oxygen and sulfide (*Figure 1*). At 100 meters a single SUP05 SAG (of 12) displayed current viral infection, while the percentage of infected SUP05 SAGs increased to 28% and 47% at 150 and 185 meters (*Figure 1— source data 1*). Consistent with previous studies evaluating community-averaged lytic viral activity (*Weinbauer et al., 2003*), cell-specific lytic viral infection estimates peaked where SUP05 is typically most abundant and metabolically active in the Saanich Inlet water column (*Hawley et al., 2014*). Additionally, remnants of past infections were detected in SUP05 and Arctic96BD-19 SAGs, including 13 putative prophages and 25 CRISPR sequences (*Supplementary file 2*). None of these 'past infection' sequences match the detected 'current infection' viral contigs.

## Patterns of co-infection between SUP05 ssDNA and dsDNA viruses

To better understand the ecological and evolutionary forces shaping SUP05 virus–host interactions in Saanich Inlet, we focused on 12 viral reference contigs including 4 *Caudovirales* contigs longer than 15 kb (from 3 *Podoviridae* and 1 *Siphoviridae*) and 8 complete genomes of *Microviridae*. Genome organization (*Figure 2*) and phylogenetic analysis (*Figure 2—figure supplement 1*) revealed that all four *Caudovirales*

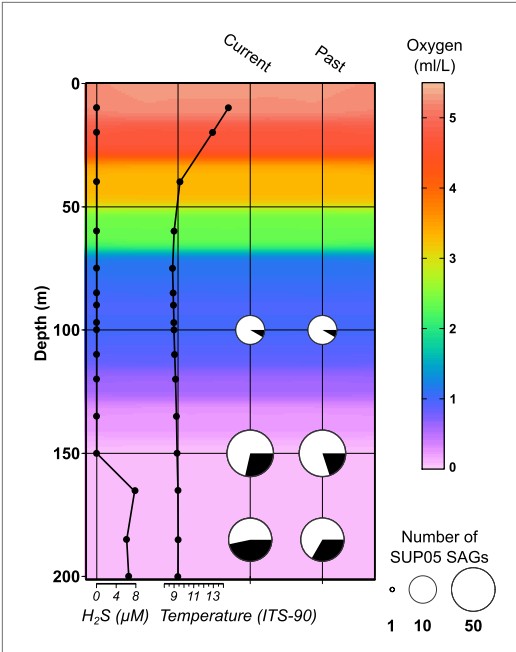

**Figure 1**. Saanich Inlet water column characteristics and SUP05 infection frequency on the SAG sampling date (August 2011). Key abiotic measurements are represented as background coloring (oxygen levels) and black lined graphs at left (hydrogen sulfide and temperature). SUP05 viral infections determined from 127 SAGs are indicated at right by black slices in pie charts where current infections were delineated from intact viral contigs and past infections were inferred from identification of defective prophages and CRISPR loci.

The following source data and figure supplements are available for figure 1:

**Source data 1**. Number of SUP05 viral sequences detected at the three different depths sampled.

**Figure supplement 1**. CTD measurements of oxygen concentration, temperature, salinity, and $H_2S$ concentration in the water column of Saanich Inlet at the time of sampling (August 2011).

**Figure supplement 2**. Phylogenetic tree of SUP05 and Arctic96BD-19 lineages based on comparative SSU ribosomal RNA gene analysis.

**Figure supplement 3**. Metrics measured on SUP05 SAG contigs classified as 'Microbial', 'Viral hallmark contigs' (**Supplementary file 1** A, B, C) and 'Putative viral contigs' (**Supplementary file 1** D).

contigs represent new genera (share <40% of their genes, *Lavigne et al., 2008*, *Figure 2—source data 1*) even when considering the viruses recently assembled from SUP05-dominated microbial metagenomes (*Anantharaman et al., 2014*). All 8 *Microviridae* contigs shared 100% nucleotide identity, despite their recovery from different SUP05_03 SAGs (**Supplementary file 3**), and represent a new genus within the subfamily *Gokushovirinae* (*Figure 2—figure supplements 2 and 3*). These identical *Microviridae* genomes could represent a lineage-specific viral bloom, targeting the SUP05_03 subclade. SUP05 infection by *Gokushovirinae* extends the known host range from small parasitic bacteria (namely *Chlamydia, Bdellovibrio* and *Spiroplasma*) to include free-living *Gammaproteobacteria*, the first marine host identified for this subfamily of viruses (*Labonté and Suttle, 2013a*).

Curiously, most (11 of 12) *Microviridae*-infected SUP05_03 SAGs also contained *Podoviridae* contigs (**Supplementary file 4**). While previously postulated based on comparative genomics, lineage-specific co-infection between the ssDNA *Microviridae* and dsDNA phages has not been observed (*Roux et al., 2012*). Such highly correlated co-occurrence in SUP05 SAGs (Fisher exact test p-value = 2e$^{-15}$) is consistent with non-random co-infection. This could be linked to cooperative infection modes between viruses or opportunistic infection of cells already infected by the other virus type, as seen in the case of satellite viruses and virophages (*Murant and Mayo, 1982*; *La Scola et al., 2008*). It is worth noting that the exact nature of interaction between satellite and helper viruses, or between virophages and their associated viruses, is still a matter of debate, and this association between two phages previously thought to be autonomous and independent (*Microviridae* and *Caudovirales*) presents a new variation on this theme (*Desnues and Raoult, 2012*; *Krupovic and Cvirkaite-Krupovic, 2012*; *Fischer, 2012*). Because the modular theory of phage evolution postulates that phage genomes consist of collections of gene modules, exchanged through proximity-enhanced recombination (*Hendrix et al., 2000*) such co-infection of a single host by ssDNA and dsDNA phages provides evidence for how such chimeric ssDNA–dsDNA viral genomes may come into existence (*Diemer and Stedman, 2012*; *Roux et al., 2013*).

## SUP05 viruses endemic to Saanich Inlet are stable over time

To extend our analysis of SUP05 virus–host interactions beyond individual SAGs, we used the 12 reference viral contigs (i.e., the 4 *Caudovirales* and 8 *Microviridae*) as platforms to recruit 3 years of Saanich Inlet microbial metagenome sequences spanning the redoxcline (*Figure 3*, **Supplementary file 5**).

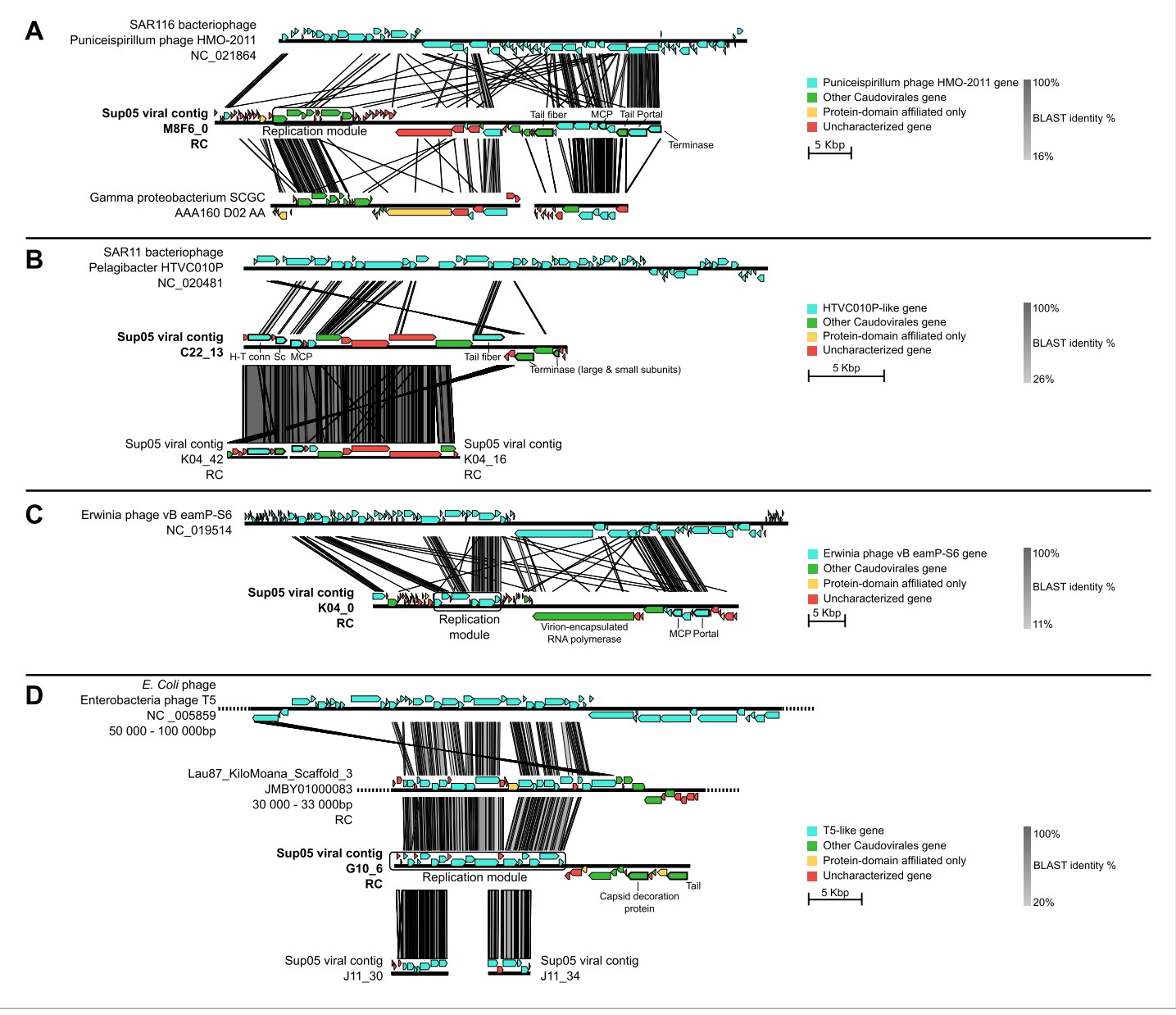

**Figure 2**. Genetic map and synteny plots for the four references SUP05 *Caudovirales* contigs M8F6_0 (**A**), C22_13 (**B**), K04_0 (**C**) and G10_6 (**D**) (high-lighted in bold). Viral hallmark genes are underlined and identified on plots (MCP: major capsid protein, Sc: scaffolding protein, H-T conn.: head-tail connector). Sequence similarities were deduced from a tBLASTx comparison. For clarity sake, several sequences including SUP05 viral contig M8F6_0, K04_0, and G10_6 are reverse-complemented (noted RC).

The following source data and figure supplements are available for figure 2:

**Source data 1**. Summary of best BLAST hit affiliation for the predicted genes of the five SUP05 reference viral contigs.

**Figure supplement 1**. Phylogenetic tree of SUP05 *Podoviridae* contigs, derived from major capsid protein sequences with PhyML (maximum-likelihood tree, LG model, CAT approximation of gamma parameter).

**Figure supplement 2**. Phylogenetic tree for the SUP05 *Microviridae* (major capsid protein).

**Figure supplement 3**. Genetic map and synteny plots for the SUP05 Microviridae reference.

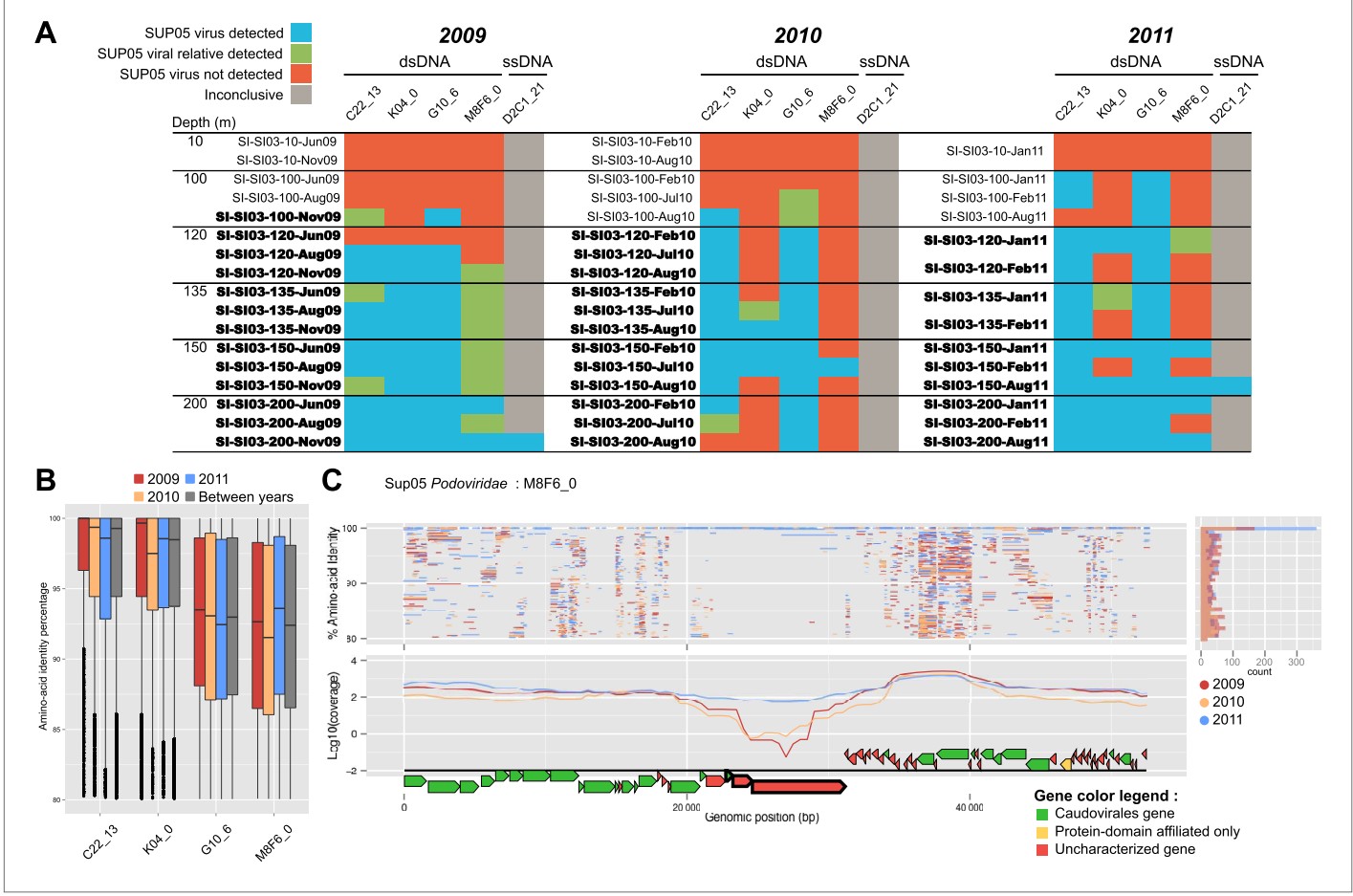

**Figure 3**. Spatiotemporal dynamics of SUP05 viral reference genomes in Saanich Inlet. (**A**) SUP05 viral presence in Saanich Inlet microbial metagenomes with OMZ sample names bolded. Four categories indicate the SUP05 virus was detected (>75% of viral genes detected at >80% amino-acid identity; light blue), a SUP05 viral relative was detected (>75% of viral genes detected at 60–80% amino-acid identity; light green), no SUP05 virus was detected (red) or detection was inconclusive (e.g., *Microviridae* in HiSeq Illumina data sets that strongly select against ssDNA sequences; gray). (**B**) SUP05 viral reference genomes had differing sequence conservation among recruited metagenomic reads. Upper and lower 'hinges' correspond to the first and third quartiles (the 25th and 75th percentiles), while outliers are displayed as points (values beyond 1.5 * Inter-Quartile Range of the hinge). (**C**) One SUP05 viral reference genome with low sequence conservation revealed evolution in action whereby a genomic region (see ~21–30 kb) appears to sweep through the population.

The following figure supplements are available for figure 3:

**Figure supplement 1**. Recruitment and coverage plot of SUP05 viral genome fragments by Saanich Inlet datasets sampled in 2009, 2010, and 2011.

**Figure supplement 2**. Heatmap of detection of SUP05 viruses in oceanic data sets.

**Figure supplement 3**. Recruitment and coverage plot of SUP05 viral genomes by data sets sampled outside of Saanich Inlet fjord.

SUP05 *Microviridae* contigs were inconsistently detected due to known methodological biases associated with linker-amplified metagenome library construction ('Materials and methods'), so we focused on dsDNA viral contigs. All 4 SUP05 *Caudovirales* contigs were absent from surface waters, but repeatedly detected within and below the oxycline, consistent with SUP05 water column disposition (*Figure 3A—figure supplement 1*). Within the *Caudovirales*, recruited microbial metagenome sequences were more similar to the reference genome for *Podoviridae* contigs C22_13 and K04_0 (96% average amino-acid identity), than for *Siphoviridae* G10_6 and *Podoviridae* M8F6_0 (92% average amino-acid identity, *Figure 2B*). Beyond sequence variation, metagenome coverage in one region of M8F6_0 (3 hypothetical open reading frames) was absent in 2009, minimal in 2010, and as

abundant as surrounding genomic regions in 2011 (*Figure 3C*), suggesting a selective sweep within this population. Contig-derived abundances of SUP05-*Caudovirales* were in sync with host distributions, but at virus-to-host ratios of 0.01 to 0.3 (*Figure 4*). While tightly choreographed virus–host abundance dynamics parallels that of cultured virus–host systems (e.g., cyanophages—[*Waterbury and Valois, 1993*]), the systematically lower (orders of magnitude lower than typical community measurements) virus-to-host ratios observed here indicates that a greater diversity of SUP05 viruses remains to be uncovered in the Saanich Inlet water column.

To determine SUP05 viral biogeography, we interrogated 74 viromes and 112 microbial metagenomes sourced from Pacific Ocean waters (*Supplementary file 5*). Despite consistently recovering SUP05 viral sequences in Saanich Inlet, these sequences were extremely uncommon in other locales (22 instances out of 803 possibilities; *Figure 3—figure supplements 2 and 3*), even when proximal to Saanich Inlet (e.g., northeastern subarctic Pacific [NESAP] coastal and open ocean waters along the LineP transect) or when sourced from similar water column conditions (e.g., Eastern Tropical South Pacific OMZ, ETSP). Of the 22 SUP05-related viruses detected, all but two were recovered below 500 meters in NESAP OMZ samples, in which SUP05 bacteria were also detected with similar abundance as in Saanich Inlet samples. The remaining two detections derived from an ETSP OMZ virome and a hydrothermal vent plume microbial metagenome from the Guaymas basin. Taken together, these observations point to endemic SUP05 viral populations with the potential to modulate SUP05-mediated biogeochemical cycling via lysis or metabolic reprogramming.

## Potential impact of SUP05 phages on sulfur metabolism

Recent studies have highlighted the role of viruses in metabolic reprogramming, from global photosynthesis (*Mann et al., 2003*; *Lindell et al., 2005*; *Clokie et al., 2006*; *Sullivan et al., 2006*; *Sharon et al., 2009*) to central carbon metabolism (*Sharon et al., 2011*; *Thompson et al., 2011*; *Hurwitz et al., 2013*) via auxiliary metabolism genes (AMGs). Additionally, viruses assembled from microbial metagenomes

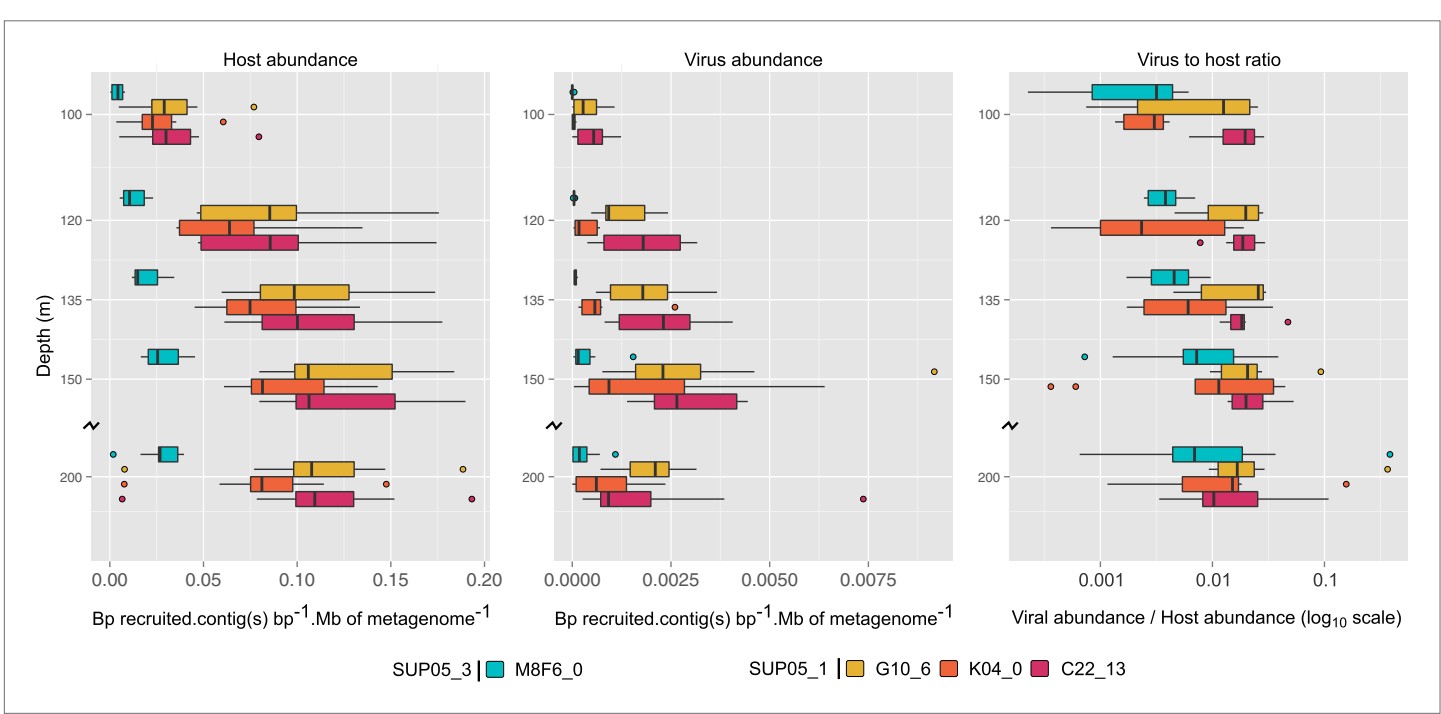

**Figure 4**. Uncultivated SUP05 lineage-specific virus–host ecology. Fragment recruitment from Saanich Inlet microbial metagenomes to microbial (95% nucleotide identity) and viral (100% amino-acid identity) reference contigs normalized by contig and metagenome size was used as a proxy for abundance. Hence, the relative abundance of microbial and viral genome is indicated as number of metagenomic bases recruited by contig(s) base pairs (bp) by megabase (Mb) of metagenome. Upper and lower 'hinges' of the relative abundance distribution correspond to the first and third quartiles (the 25th and 75th percentiles), while outliers are displayed as points (values beyond 1.5 * Inter-Quartile Range of the hinge). A virus-to-host ratio was then calculated for each SAG (i.e., each virus-host pair) as the ratio of relative abundance of viral contigs to the relative abundance of microbial contigs from the same SAG.

from SUP05 dominated hydrothermal vent samples contain sulfur cycling genes (*Anantharaman et al., 2014*). Therefore, we looked for AMGs encoded on SUP05 viral contigs in the Saanich Inlet water column.

Four putative AMGs were detected in 12 of the 69 viral contigs, predominantly from SUP05_01 SAGs recovered from 150 meters (*Supplementary file 6*). One AMG identified on a *bona fide* viral contig, phosphate-related *phoH*, is common among marine phages, but remains functionally uncharacterized (*Sullivan et al., 2010*; *Goldsmith et al., 2011*). The remaining 3 AMGs including 2-oxoglutarate (2OG) and Fe(II)-dependent oxygenase superfamily (2OG-FeII oxygenase), tripartite tricarboxylate transporter (*tctA*, protein domain hit only), and dissimilatory sulfite reductase subunit C (*dsrC*) were encoded on contigs identified by non-reference-based methods. In marine cyanophages, 2OG-FeII oxygenase-encoding genes are common where they are thought to modulate host nitrogen metabolism during infection (*Sullivan et al., 2010*). However, the precise metabolic role of *tctA* and *dsrC*-like genes during viral infection remains unknown.

Given that *dsrC* was found on 7 SUP05_01 viral contigs (*Supplementary file 7*) and DsrC is critical in SUP05 energy metabolism (*Walsh et al., 2009*), we focused on this gene. Although *dsrC* genes were only present on contigs identified by non-reference-based methods they were closely related to *dsrC*-like genes encoded on the hydrothermal vent plume phages (*Anantharaman et al., 2014*). Indeed, conceptually translated sequence alignment of these viral *dsrC* genes including putative viral and bacterial genes from microbial metagenomic data sets indicate that the Saanich Inlet 'viral' sequences belong to one *dsrC* subgroup (*dsrC_1* according to the classification of *Anantharaman et al., 2014*). In addition to high sequence similarity viral *dsrC* genes from SUP05 SAGs co-localized on contigs with viral homologs (e.g., 2OG-FeII oxygenase, chaperonin), and occurred in genomic context that was completely different to the conserved and well-characterized *dsrC* region in SUP05 genomes (*Figure 5A,B*).

The *dsrC_1* group encodes a protein retaining 15 conserved residues across known DsrC subunits. However, the second C-terminal cysteine and a 7–8 residue insertion thought to be required for DsrC function based on structural analysis of *Desulfovibrio vulgaris* and *Archaeoglobus fulgidus* proteins are missing from the viral protein (*Figure 5—figure supplement 1*; *Mander et al., 2005*; *Oliveira et al., 2008*). These differences suggest that either the viral encoded *dsrC* is non-functional or has a modified function. Given that genes shared between different viral genomes rarely represent nonfunctional genes, it is likely that viral-encoded *dsrC* plays a biological role in SUP05. Indeed, there is precedent for divergent viral AMGs serving as modified functional counterparts to host-encoded homologues. Specifically, a highly divergent viral '*pebA*' (*Sullivan et al., 2005*) was experimentally demonstrated to perform the functions of two host enzymes' (*pebA* and *pebB)* as a bifunctional enzyme, phycoerythrobilin synthetase (*pebS*) (*Dammeyer et al., 2008*).

Given that viral *dsrC* genes were abundant in the Saanich Inlet water column over a 3-year-time interval (*Figure 5C*) with peaked recovery consistent with blooming SUP05 populations (*Figure 5—figure supplement 2*; *Hawley et al., 2014*), we posit that this viral gene is functional in SUP05 sulfur cycling. Future functional characterization of viral DsrC is needed to constrain viral roles in modulating SUP05 electron transfer reactions during viral infection in the environment.

## Conclusion

While new methods and model systems for identifying virus–host interactions continue to emerge (*Tadmor et al., 2011*; *Allers et al., 2013*; *Mizuno et al., 2013*; *Deng et al., 2014*), viral ecology remains predominantly community focused in nature. This is because most hosts are uncultivated (*Rappé and Giovannoni, 2003*), and culture-independent viral metagenomes are dominated by 'unknown' sequences (*Hurwitz and Sullivan, 2013*), which inhibits developing a mechanism- and population-based viral ecology. Here, we use single-cell genomics to directly link SUP05 viruses and their hosts across defined gradients of oxygen and sulfide over a 3-year-time interval in a model OMZ ecosystem. This spatiotemporal resolution revealed endemic patterns of co-infection between ssDNA and dsDNA viruses and the occurrence of AMGs with the potential to modulate electron transfer reactions essential to SUP05 energy metabolism. Together, these findings offer novel perspectives on the ecology and evolution of viruses infecting uncultivated bacterial populations. While the capacity to formulate such linkages between cultured virus–host systems in nature is recognized (e.g., cyanophages and pelagiphages), the use of single-cell genomics to explore such linkages in uncultivated microbial communities represents a watershed moment in illuminating viral dark matter and its role in modulating microbial interaction networks in natural and engineered ecosystems.

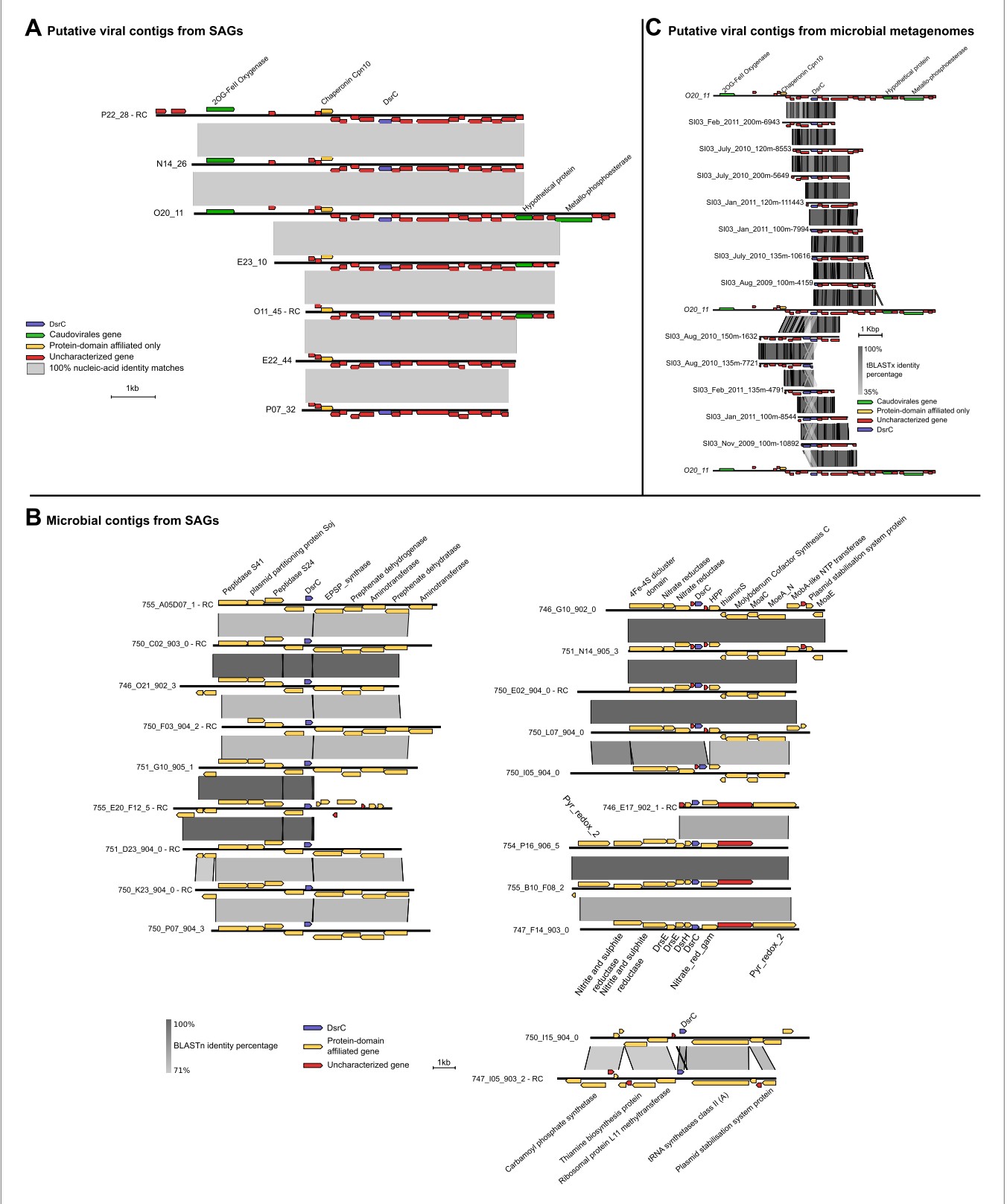

**Figure 5**. Maps of DsrC-containing contigs. (**A**) Seven contigs including *dsrC*-like gene detected as viral based on non-reference metrics (ratio of uncharacterized genes, strand coding bias). (**B**) Genomic context in which *dsrC*-like genes are retrieved in SUP05 microbial contigs from SAG. All contigs above 50 kb containing a *dsrC*-like gene were selected and compared to get a summary of the different regions in which *dsrC*-like genes are found in *Figure 5. Continued on next page*

*Figure 5. Continued*

SUP05 genomes. (**C**) Map of *dsrC*-containing Contigs assembled from Saanich Inlet metagenomes. One viral-like contig from SAG (020_11) is included for comparison.

The following figure supplements are available for figure 5:

**Figure supplement 1**. Multiple alignment of *dsrC*-like genes from Saanich Inlet microbial and viral contigs, hydrothermal vent phages, and microbial genomes.

**Figure supplement 2**. Relative abundance of viral dsrC gene on the 3 years of sampling in Saanich Inlet compared to the concentration of $H_2S$ (left) and $O_2$ (right).

## Materials and methods

### Sample collection, sequencing, and assembly

Samples were collected in Saanich Inlet on Vancouver Island, British Columbia, on the 09th of August 2011. Sample collection and biochemical measurements were performed as previously described (*Zaikova et al., 2010*). Water column redox conditions were typical for stratified summer months when SUP05 populations bloom in deep basin waters. Individual cells <10 µm in diameter from 100, 150, and 185 meter depth samples were subjected to fluorescence-activated cell sorting, multiple displacement amplification (MDA), and taxonomic identification at the Bigelow Laboratory Single Cell Genomics Center (SCGC; http://scgc.bigelow.org), following previously described procedures (*Stepanauskas and Sieracki, 2007*; *Swan et al., 2013*). A total of 315 single amplified genomes (SAGs) per sample were subjected to multiple displacement amplification (MDA), and the taxonomic identity of single amplified genomes (SAG) was determined by directly sequencing bacterial small subunit ribosomal RNA (SSU rRNA) gene amplicons. A total of 136 SAGs affiliated with SUP05 or Arctic96BD-19 were selected for genome sequencing. Between 1 and 3 µg of MDA product was sent to Canada's Michael Smith Genome Sciences Center (Vancouver, BC) to create shotgun libraries. Briefly, the DNA was sheared to 350–450 bp fragments using a Covaris E210 and purified using AMPure XP Beads according to the manufacturer's instructions. The sheared DNA was end-repaired and A-tailed according to the Illumina standard PE protocol and purified again using AMPure XP Beads, generating paired-end 100-bp reads. Indexed libraries were amplified by PCR for six cycles, gel-purified, pooled (11–12 samples per lane), and QC assessed on a Bioanalyzer DNA Series II High Sensitivity chip (Agilent, Santa Clara, CA, USA), and then sequenced using an Illumina HiSeq2000 sequencer.

All raw Illumina sequence data were passed through DUK, a filtering program developed at JGI, which removes known Illumina sequencing and library preparation artifacts (Mingkun, Copeland, and Han, Unpublished). Artifact filtered sequence data were then screened and trimmed according to the k-mers present in the data set (Mingkun and Kmernorm, Unpublished). High-depth k-mers, presumably derived from MDA amplification bias, cause problems in the assembly, especially if the k-mer depth varies in orders of magnitude for different regions of the genome. Reads with high k-mer coverage (>30× average k–mer depth) were normalized to an average depth of 30×. Reads with an average k-mer depth of less than 2× were removed. Following steps were then performed for assembly: (i) normalized Illumina reads were assembled using IDBA–UD version 1.0.9 (*Peng et al., 2012*); (ii) 1–3 kb simulated paired end reads were created from IDBA–UD contigs using wgsim (https://github.com/lh3/wgsim); (iii) normalized Illumina reads were assembled with simulated read pairs using Allpaths–LG (version r42328) (*Gnerre et al., 2011*); (iv) Parameters for assembly steps were: (i) IDBA–UD (—no local), (ii) wgsim (–e 0 –1 100 –2 100 –r 0 –R 0 –X 0), (iii) Allpaths–LG (PrepareAllpathsInputs: PHRED 64=1 PLOIDY=1 FRAG COVERAGE=125 JUMP COVERAGE=25 LONG JUMP COV=50, RunAllpathsLG: THREADS=8 RUN=std shredpairs TARGETS=standard VAPI WARN ONLY=True OVERWRITE=True MIN CONTIG=2000).

### SAG taxonomic assignment

SAG taxonomy was verified using the assembled contigs in two ways using MetaPathways 1.0 (*Konwar et al., 2013*). First, the assemblies were blasted against the SILVA (v.111) database to confirm the taxonomy based on SSU rRNA. Next, MEGAN5 was used to carry out taxonomic binning of all ORFs from the MetaPathways BLAST output using the Lowest Common Ancestor (LCA) approach (*Huson et al., 2007*).

A total of 2711 SSU rRNA sequences previously taxonomically assigned to SUP05 and Arctic96BD-19 lineages were aligned and clustered using mothur v.1.27.0 (*Schloss et al., 2009*), and 20 representative sequences for the most abundant clusters (cutoff = 6) at 97% similarity were selected. These representative sequences were used to build the phylogenetic tree differentiating between SUP05 and Arctic96BD-19. Reference SUP05 and Arctic96BD-19 sequences from different environments and symbionts and cluster representative sequences were aligned using the SILVA aligner tool (http://www.arb-silva.de/aligner/) and imported into an in-house ARB database for SUP05. Aligned sequences were exported from ARB into Mesquite for manual alignment refinement. The final phylogenetic tree was inferred from manually refined Mesquite alignment of sequences using maximum likelihood implemented in PHYML using a GTR model with estimated values for the α parameter of the Γ distribution and the proportion of invariable sites. The confidence of each node was determined by assembling a consensus tree of 1000 bootstrap replicates.

## Microbial and viral metagenomes

The protocols used to generate the POV (*Hurwitz and Sullivan, 2013*), ETSP OMZ viromes (*Cassman et al., 2012*), ETSP microbial metagenomes and metatranscriptomes (*Stewart et al., 2012*; *Ganesh et al., 2014*), and Guaymas basin metagenome (*Anantharaman et al., 2013*) are described in their respective publications. All these data sets were sequenced with Roche 454 GL FLX Titanium systems, and quality controlled reads were used in the different analysis computed in this study.

LineP and Malaspina viral metagenomes (viromes) were obtained from samples collected during LineP (http://www.pac.dfo-mpo.gc.ca/science/oceans/data-donnees/line-p/index-eng.html) and Malaspina (http://scientific.expedicionmalaspina.es/) cruises. Particles were precipitated with Iron–Chloride from 0.2 µm filtrates, and resuspended in EDTA-Mg-Ascorbate buffer (*John et al., 2011*) before the DNA was extracted using Promega's Wizard Prep kit. Assembly and gene prediction were conducted through the IMG/M ER pipeline (*Markowitz et al., 2014*). Microbial metagenome samples at Saanich Inlet and along the LineP transect were also collected during LineP cruises (http://www.pac.dfo-mpo.gc.ca/science/oceans/data-donnees/line-p/index-eng.html). Sequencing and assembly of these data sets was conducted at the JGI. A list of the different web servers and accession numbers for these publicly available data sets is displayed in *Supplementary file 5*.

## Detection of viral contigs in SUP05/Arctic SAG

SUP05 SAG contigs were annotated with the Metavir web server (*Roux et al., 2014*). Briefly, ORFs were predicted with MetaGeneAnnotator (*Noguchi et al., 2008*) and compared to the RefseqVirus database with BLASTp (*Altschul et al., 1997*). In order to select viral-associated contigs, we looked for viral-specific genes, that is, genes associated with the formation of the capsid and encapsidation of the genome (designated as 'hallmark viral genes'). Thus, we searched for all genes annotated as 'virion structure', 'capsid', 'portal', 'tail', or 'terminase', and selected contigs including at least one of these hallmark genes (*Supplementary file 1*). Among the 50 viral contigs detected, we highlighted a set of 12 long (>15 kb) or circular contigs as the best references available for SUP05 phages (*Supplementary file 1*). We then compared the reference sequences retrieved in this first screening round to all the SUP05/Arctic96BD-19 SAG contigs, in order to extract more viral-related sequences (*Supplementary file 1*). At this step, all contigs with at least 50% of their genes similar to a previously detected SUP05 viral contigs were retained (sequence similarity between predicted genes assessed through BLASTp, thresholds of 0.001 for e-value and 50 for bit score).

Alternatively, we compared the SUP05/Arctic96BD-19 SAG contigs to a set of ocean viromes (*Supplementary file 5*) and looked for every contig which was covered by virome reads (for 454-sequenced viromes) or predicted genes (for HiSeq-sequenced viromes) on at least three genes with at least 90% of identity (protein sequences). However, this comparison to viromes only highlighted contigs already identified as viral from the hallmark gene analysis. Finally, we looked for every sequence which could come from a new type of phage, based on two known properties of phage genomes: most of their genes are not similar to anything in the current databases, and they tend to be mostly coded on the same strand (by block, or module) (*Akhter et al., 2012*). We thus looked for all regions in SAG contigs composed of at least 50% of uncharacterized genes, with at least 80% of them on the same coding strand. 19 new short viral contigs were highlighted through this detection (*Supplementary file 1*), which displayed characteristics close to the viral hallmark contigs (*Figure 1—figure supplement 3*).

A set of regions of putative viral origin within bacterial contigs also stood out. These sequences were manually curated to check if they could indeed be of viral origin, notably by checking if these regions were conserved between closely related bacterial contigs, and 13 putative defective prophages were eventually identified among them. CRISPR regions were detected with the CRISPR recognition tool (*Bland et al., 2007*). All spacers were extracted and compared to all SUP05/Arctic96BD-19 SAG contigs with BLASTn.

## Annotation of viral contigs

The annotations of selected contigs were extracted from the Metavir web server (*Roux et al., 2014*) and manually curated. Taxonomic affiliations were based on a BLAST comparison to RefseqVirus and NR databases from NCBI, with a bit score threshold of 50 and e-value threshold of 0.001. A tBLASTx comparison of larger contigs (>15 kb) against WGS (Whole-Genome shotgun), HTGS (High-Throughput Genomic Shotgun), and GSS (Genomic Survey Sequences) from the NCBI was used to add the most closely related sequence to the analysis, which could have not been included in the NR and Refseq database yet. This screening notably lead to the detection of two contigs from a Gammaproteobacteria single-cell amplified genome (Gamma proteobacterium SCGC AAA160-D02) similar to SUP05 phage genome and was therefore included in the phylogenetic and genome comparison analysis. The affiliation of SUP05 viruses to new or existing genera was based on the criteria of 40% of genes shared within a genus previously defined for *Caudovirales* (*Lavigne et al., 2008*). Map comparison figures were created with Easyfig (*Sullivan et al., 2011*).

Functional annotation was achieved through a domain search against the PFAM database (*Punta et al., 2012*) (hmmscan [*Eddy, 2011*], using a threshold of 0.001 for e-value and 30 for score). When looking for putative AMGs, defective prophages were not considered since these regions are likely to be subject to rearrangement and gene transfer, and the origin of single genes within these regions is uncertain. A set of microbial *dsrC* sequences were selected as references for SUP05 viral-encoded *dsrC* genes in genomic context (*Figure 5B*). Briefly, all contigs in SUP05 SAGs longer than 50 kb and containing a DsrC-like gene were compared through BLASTn and displayed with Easyfig (*Sullivan et al., 2011*).

## Phage multiple alignments and phylogenetic trees

Maximum-likelihood trees were computed with PhyML (*Guindon and Gascuel, 2003*) using a LG model, a CAT approximation for Gamma parameter, and computing SH-like scores for node supports. All SUP05 contigs affiliated to *Podoviridae* and including the major capsid protein gene were added in a single tree alongside reference sequences from *Autographivirinae* and N4-like viruses. The most closely related sequences to each SUP05 *Podoviridae*, as detected from the genome comparison analysis, were also included in the tree. SUP05 *Microviridae* were included in a phylogenetic tree based on the Major Capsid protein and centered around the *Gokushovirinae* sub-family, with sequences from *Pichovirinae* used as outgroup. *Gokushovirinae* reference sequences were taken from *Roux et al. (2012)* and *Labonté and Suttle (2013b)*. In order to include more aquatic sequences, complete *Microviridae* genomes were assembled from two sets of viromes sampled from a freshwater subtropical reservoir (*Tseng et al., 2013*) and deep-sea sediments (*Yoshida et al., 2013*) and annotated as previously described (*Roux et al., 2012*). Tree figures were drawn with Itol (*Letunic and Bork, 2007*). DsrC-like predicted protein sequences were aligned with Muscle v3.8.31 (*Edgar, 2004*), and the multiple alignment was displayed with Jalview (*Waterhouse et al., 2009*).

## Recruitment of metagenomic sequences to SUP05 viral genomes

A set of oceanic viromes and microbial metagenomes were used for comparison with SUP05 viral genomes (*Supplementary file 5*). Similarities between SUP05 viral genomes and published viromes were assessed through BLAST comparison, BLASTx for 454-sequenced viromes (POV data set [*Hurwitz and Sullivan, 2013*], ETSP OMZ viromes [*Cassman et al., 2012*], ETSP microbial metagenomes and metatranscriptomes [*Ganesh et al., 2014*; *Stewart et al., 2012*], and Guaymas basin metagenome [*Anantharaman et al., 2013*]) and BLASTp from predicted protein for HiSeq-sequenced viromes (LineP and Malaspina viromes, Saanich Inlet and LineP microbial metagenomes), with similar thresholds of 0.001 for e-value and 50 for bit score. Each metagenome—viral genome association was classified based on the number of viral genes detected and the amino-acid percentage identity of the BLAST hits associated: when more than 75% of the genes were detected at more than 80% identity in the metagenome, the viral genome was thought to be in the sample. The same ratio of genes detected at

lower percentage (60 to 80%) indicates the presence of a related but distinct virus. We considered that less than 75% of the genes detected meant that this virus was likely absent from the sample. The results of *Microviridae* detection with the HiSeq Illumina data sets have to be carefully considered, as the linker amplification used in the preparation of samples for HiSeq Illumina sequencing displays a strong bias against ssDNA templates such as *Microviridae* genomes (*Kim and Bae, 2011*). Hence, if the detection of SUP05 *Microviridae* in HiSeq Illumina data sets undoubtedly testifies for the presence of these viruses in the samples, an absence of detection is not a strong indicator of their absence in the sample.

In order to detect the host of SUP05 viruses in the same data sets, a mapping of all sequences from each metagenome to non-viral SAG contigs was computed with mummer (*Delcher et al., 2003*) (minimum cluster length of 100, maximum gap between two matches in a cluster of 500). The Saanich Inlet SUP05 bacteria is considered present in the metagenome when more than 75% of genes are covered by metagenomic sequences with average nucleotide identity above 95%. Viral-encoded *dsrC* was computed with a threshold of 95% on average nucleotide identity, as no similarity beyond 80% average nucleotide identity was detected between viral and microbial homologues, whether from public database or from the SUP05 SAG microbial contigs. All recruitment and coverage plots were drawn with the ggplot2 module of R software (*Wickham, 2009*).

## Abundance and variability of SUP05 viral and microbial genomes

Assessment of variability in the populations associated with each SUP05 virus was based on a BLASTp between all sequences from Saanich Inlet metagenomes recruited by each SUP05 viral contig (thresholds of 50 for bit score, 0.001 for e-value, and 80% for amino-acid identity). The relative abundance of SUP05 viral and microbial genomes was assessed from the recruitment of Saanich Inlet metagenomic reads to each viral contig and set of microbial contigs (all contigs greater than 5 kb and not identified as viral) for each 'reference' SAG (i.e., the 4 SAG in which a SUP05 reference *Caudovirales* was detected: AB-750C22AB-904 for C22_13, AB-750K04AB-904 for K04_0, AB-751_G10AB-905 for G10_6, and AB-755_M08F06 for M8F6_0, *Figure 2—source data 1*). For each metagenome, a normalized ratio of nucleotides recruited by each contig or set of contigs was calculated as the number of bases recruited (sum of the length of recruited reads) divided by the total number of bases in the (set of) contig(s) and the total number of bases in the metagenome. The ratio of viral genomes to host genomes was then calculated for each metagenome as the relative abundance of viral contig divided by the relative abundance of bacterial contig from the same SAG. The plots of genetic variability and relative abundance distributions were generated with the ggplot2 module of R software (*Wickham, 2009*). The perl scripts used in the different part of the bioinformatics analyses are available online at http://tmpl.arizona.edu/dokuwiki/doku.php?id=bioinformatics:scripts:sup05 and as *Source code 1*.

## Acknowledgements

We thank the crew aboard the MSV John Strickland for logistical and sampling support in Saanich Inlet and Melanie Scofield, Jody Wright, Evan Durno, and Elena Zaikova in the Hallam lab for technical assistance. We also thank the Joint Genome Institute, including IMG and GOLD teams and Sussanah Tringe, Stephanie Malfatti, and Tijana Glavina del Rio for technical and project management assistance. This work was performed under the auspices of the U.S. Department of Energy Joint Genome Institute supported by the Office of Science of the U.S. Department of Energy under Contract No. DE-AC02-05CH11231; the G Unger Vetlesen and Ambrose Monell Foundations and the Tula Foundation funded Centre for Microbial Diversity and Evolution, Natural Sciences and Engineering Research Council (NSERC) of Canada, Canada Foundation for Innovation (CFI), and the Canadian Institute for Advanced Research (CIFAR) through grants awarded to SJH; and BIO5, NSF (OCE-0961947) and the Gordon and Betty Moore Foundation (#3790) through grants awarded to MBS. This work was supported by the University of Arizona, Technology and Research Initiative Fund, through the Water, Environmental and Energy Solutions Initiative. Single cell genomics instrumentation at Bigelow Laboratory for Ocean Sciences was supported by NSF grants OCE-821374 and OCE-1019242 to RS and by the State of Maine Technology Institute. The single cell genome sequences and annotations can be accessed via IMG (img.jgi.doe.gov, SAG Ids are listed in *Supplementary file 4*). Viral contigs and defective prophages identified in the SUP05 SAG are available on the Metavir webserver (http://metavir-meb.univ-bpclermont.fr/), as virome 'SUP05_viral_sequences' in project 'SUP05_SAGs'. The web servers hosting viral and microbial metagenome sequences used here are listed in *Supplementary file 5*.

## Additional information

### Funding

| Funder | Grant reference number | Author |
|---|---|---|
| Office of Science | DE-AC02-05CH1123 | Ramunas Stepanauskas, Tanja Woyke, Steven J Hallam, Matthew B Sullivan |
| Ambrose Monell Foundation | | Steven J Hallam |
| Tula Foundation | | Steven J Hallam |
| Canadian Network for Research and Innovation in Machining Technology, Natural Sciences and Engineering Research Council of Canada | | Steven J Hallam |
| Canada Foundation for Innovation | | Steven J Hallam |
| Canadian Institute for Advanced Research | | Steven J Hallam |
| Gordon and Betty Moore Foundation | 3790 | Matthew B Sullivan |
| National Science Foundation | OCE-0961947 | Matthew B Sullivan |
| Bio5 Institute | | Matthew B Sullivan |
| G. Unger Vetlesen Foundation and Ambrose Monell Foundation | | Steven J Hallam |
| University of Arizona, Technology and Research Initiative Fund | Water, Environmental and Energy Solutions Inititative | Matthew B Sullivan |
| National Science Foundation | OCE-821374, OCE-1019242 | Ramunas Stepanauskas |

The funders had no role in study design, data collection and interpretation, or the decision to submit the work for publication.

### Author contributions

SR, MBS, Conception and design, Analysis and interpretation of data, Drafting or revising the article; AKH, MTB, Acquisition of data, Analysis and interpretation of data; MS, Acquisition of data, Drafting or revising the article; PS, Acquisition of data; RS, TW, Conception and design, Acquisition of data, Drafting or revising the article; SJH, Conception and design, Acquisition of data, Analysis and interpretation of data, Drafting or revising the article

## Additional files

### Supplementary files

• Supplementary file 1. List of viral sequences and defective prophages retrieved in SUP05/Arctic SAGs. Upper part of the table displays the 12 'SUP05 viral reference' sequences detected from the presence of viral hallmark gene and their size greater than 15 kb or circularity (**A**), then the 19 SUP05 short viral contigs (**B**), with taxonomic affiliation based on viral hallmark genes. The bottom part displays the 19 other sequences retrieved through the second screening (**C**), based on the first set as references (including contigs previously detected as 'SUP05 short viral contigs'), and the 18 other putative viral contigs (**D**), which affiliation to the viral kingdom is uncertain since they lack a viral hallmark gene. Estimated genome sizes are based on the size of the most closely related phage genomes, or in the case of the *Microviridae* on the length of the circular contigs.

• Supplementary file 2. List of contigs containing a putative defective prophage (A) or a CRISPR locus (B).

• Supplementary file 3. Number of genes shared between contigs of Single-Amplified Genome (SAG) with a Gokushovirinae genome and the contigs of the five most closely related SAGs. For each SAG with a *Gokushovirinae* genome, the five SAGs displaying the most identical genes (100% amino-acid identity) are indicated. The number and ratio of identical genes is displayed for each pair of SAGs, alongside the number and ratio of genes similar but non identical (BLASTp hit with bit score greater than 50, e-value lower than 0.001, and identity percentage greater than 30%). Matching SAGs which also display a *Gokushovirinae* genome are noted with a star.

• Supplementary file 4. List of viral sequences detected for each SUP05 SAGs with at least one viral contig or defective prophage. For the detection of viral contigs, full-length contigs are indicated by a cross (x), partial matches (short contigs matching the full-length sequence) are noted with a dash (−). For the short contigs not similar to any SUP05 viral reference sequence, the number of different contigs identified is indicated for each cell.

• Supplementary file 5. List of metagenomic data sets used in this study. Viral metagenomes were used for both viral contig detection and recruitment plots, whereas microbial metagenomes were only included in the recruitment plot computation. OMZ samples are highlighted in bold.

• Supplementary file 6. List of PFAM domains detected in the 68 viral sequences identified. The four putative Auxiliary Metabolism Genes are highlighted in bold.

• Supplementary file 7. Number of genes shared between contigs of Single-Amplified Genome (SAG) with a DsrC gene on a viral contig and the contigs of the five most closely related SAGs. For each SAG with a DsrC gene on a viral contig, the five SAGs displaying the most identical genes (100% amino-acid identity) are indicated. The number and ratio of identical genes is displayed for each pair of SAGs, alongside the number and ratio of genes similar but non identical (BLASTp hit with bit score greater than 50, e-value lower than 0.001, and identity percentage greater than 30%). Matching SAGs which also display a similar DsrC gene on a viral contig are indicated with a star.

• Source code 1. Set of perl scripts used to (i) evaluate metrics (gene size, strand bias, ratio of uncharacterized genes) and detect phage sequences in the SAG dataset, (ii) compute relative abundance of phages and hosts and generate recruitment plots from BLAST comparison of metagenomes and SAG contigs, and (iii) evaluate the genetic diversity within reads recruited to a phage contig.

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
