## [Decision Letter]

Thank you for sending your work entitled “Cultivation-independent exploration of SUP05 virus-host interactions in a model Oxygen Minimum Zone” for consideration at *eLife.* Your article has been favorably evaluated by Ian Baldwin (Senior editor) and 3 reviewers, one of whom is a guest Reviewing editor.

The Reviewing editor and the other reviewers discussed their comments before we reached this decision, and the Reviewing editor has assembled the following comments to help you prepare a revised submission.

All 3 reviewers agree that Roux et al. present a fascinating, multifaceted study of high quality that analyses the viral community of a group of as yet uncultivable bacteria called SUP05 using single-cell amplified genome (SAG) sequencing. SUP05 are highly abundant in marine oxygen minimum zones (OMZ), areas of the ocean that are expanding and associated with the generation of greenhouse gases. Given that viruses can have a major effect on microbially driven carbon, nitrogen and sulfur transformations, improving our knowledge of viruses associated with SUP05 is critical for a better understanding of OMZs and their effects on marine biogeochemical processes.

Roux et al sequenced SUP05 cells from three different water depths across the chemocline, allowing them to draw inferences about infection rates as a function of water column geochemistry. Major results include the first estimate for viral infection rates in free-living bacteria from the environment (as high as 33%), host-specific identification of viruses in an environmental sample, co-infection of hosts by ssDNA and dsDNA viruses, the presence of auxiliary metabolic genes including dsrC (involved in sulfur oxidation), and increases in viral infection frequencies with water depth and oxygen deficiency.

We have the following suggestions for revising the paper:

1) Since this paper was submitted, a paper describing viruses that infect SUP05 in deep-sea hydrothermal plumes was published by Anantharaman et al. (Science 344: 757). These two papers are quite distinct in their approach, field site, and results, so the Anantharaman et al. study does not diminish the impact of the current manuscript. However, both the common themes (auxiliary metabolic genes (dsrC); some shared taxa) as well as differences (no dsrA in this paper; some differences in taxa identified) should be discussed.

2) The paper provides a large quantity of data and covers the topic well; however a more general discussion and a little less method in the main text region is needed. There is no attempt to compare the findings to other viral data from OMZ's or marine environments in general.

3) The weakest part of this paper is the section on auxiliary metabolism genes (AMGs). While this is only a side aspect of this paper, it is important that it be addressed. In both this submission and the Anantharaman et al. paper, the annotation of the DsrC protein is problematic: DsrC is a short protein and has homologs that are very likely not involved in sulfur oxidation by rDSR or sulfur reduction by DSR, such as TusE, which is involved in thiouridine biosynthesis. Anantharaman et al. identified two groups of rDsrC, but only one (their “rdsrC2 group”) has the residues that are most likely needed for the protein to function as rDsrC. The sequences in their second group (“rdsrC1”) do not share the seemingly characteristic signatures with any of the proteins TusE, rDsrC, and DsrC (“rdsrC1” sequences lack both the 2nd c-terminal cysteine and the 7-8 residue insertion). The neighboring genes for the first three sequences from the second group (“rdsrC1”) do not have any (r)DSR-related genes. Instead, there is an annotated ”thiamine biosynthesis gene”. The DsrC homologs of the ”rdsrC1 group” might therefore be homologs with a completely different function.

We would therefore like the authors to reanalyze the DsrC genes they found, e.g. by aligning them with bona fide DsrC genes and those from the Anantharaman et al. study. This is particularly interesting because the “viral” DsrC apparently occurred in a completely different genomic context than the DsrC of the host and seems to be “relatively divergent from the host version”.

4) We were not convinced by the authors' statements that they were able to show “viral metabolic reprogramming”, “virus-host co-evolution dynamics”, and reactions that “fuel SUP05-mediated nitrogen loss and inorganic carbon fixation pathways with resulting feedback on climate active trace gas cycling in OMZ waters with real world implications for biogeochemical models.” Evidence for metabolic reprogramming (such as expression of the relevant proteins and physiological data showing the effects of reprogramming), co-evolution (e.g. phylogenetic analyses and statistical tests for co-evolution), and effects of SUP05 phages on nitrogen and carbon cycles is needed to provide support for these unnecessary inflationary statements.

---

## [Author Response]

*1) Since this paper was submitted, a paper describing viruses that infect SUP05 in deep-sea hydrothermal plumes was published by Anantharaman et al. (Science 344: 757). These two papers are quite distinct in their approach, field site, and results, so the Anantharaman et al. study does not diminish the impact of the current manuscript. However, both the common themes (auxiliary metabolic genes (dsrC); some shared taxa) as well as differences (no dsrA in this paper; some differences in taxa identified) should be discussed*.

We re-analyzed our contigs and dsrC-like genes in light of the Anantharaman et al. manuscript, and added the relevant comparisons through the manuscript (Figure 2—figure supplement 1 and Figure 5—figure supplement 1). Overall, phylogenetic and synteny analysis revealed that our SUP05 phages are quite different from the one highlighted in the Anantharaman manuscript (probably due to the fact that hydrothermal vent SUP05 are distinct from OMZ SUP05). As the reviewers pointed out, the most informative comparative analysis concerned the dsrC AMG (detailed in response to question 3).

*2) The paper provides a large quantity of data and covers the topic well; however a more general discussion and a little less method in the main text region is needed. There is no attempt to compare the findings to other viral data from OMZ's or marine environments in general*.

We added a paragraph of more general introduction about other OMZ's and marine viral communities, and our results were put into the context of known OMZ viruses. We found that, in accordance with previous metagenomics studies, OMZ viruses are clearly distinct from other marine viral communities, notable from the surrounding surface and deep-sea waters. However, our SUP05 Microviridae are the first example of this family in an OMZ.

*3) The weakest part of this paper is the section on auxiliary metabolism genes (AMGs). While this is only a side aspect of this paper, it is important that it be addressed. In both this submission and the Anantharaman et al. paper, the annotation of the DsrC protein is problematic: DsrC is a short protein and has homologs that are very likely not involved in sulfur oxidation by rDSR or sulfur reduction by DSR, such as TusE, which is involved in thiouridine biosynthesis. Anantharaman et al. identified two groups of rDsrC, but only one (their ”rdsrC2 group”) has the residues that are most likely needed for the protein to function as rDsrC. The sequences in their second group (“rdsrC1”) do not share the seemingly characteristic signatures with any of the proteins TusE, rDsrC, and DsrC (”rdsrC1” sequences lack both the 2nd c-terminal cysteine and the 7-8 residue insertion). The neighboring genes for the first three sequences from the second group (“rdsrC1”) do not have any (r)DSR-related genes. Instead, there is an annotated “thiamine biosynthesis gene”. The DsrC homologs of the “rdsrC1 group” might therefore be homologs with a completely different function*.

*We would therefore like the authors to reanalyze the DsrC genes they found, e.g. by aligning them with bona fide DsrC genes and those from the Anantharaman et al. study. This is particularly interesting because the ”viral” DsrC apparently occurred in a completely different genomic context than the DsrC of the host and seems to be “relatively divergent from the host version”*.

The DsrC genes were re-analyzed and compared to both microbial sequences and the new viral DsrC from the Anantharaman et al. study. This lead to a new supplemental figure (Figure 5—figure supplement 1), and an addition to the AMG paragraph. We found out that the dsrC genes in our OMZ SUP05 phages corresponded to one of the two categories described in the Anantharaman et al., category with an incomplete set of conserved residues which suggest a modified role for these genes (beyond sulfur reduction).

*4) We were not convinced by the authors' statements that they were able to show “viral metabolic reprogramming”, “virus-host co-evolution dynamics”, and reactions that “fuel SUP05-mediated nitrogen loss and inorganic carbon fixation pathways with resulting feedback on climate active trace gas cycling in OMZ waters with real world implications for biogeochemical models.” Evidence for metabolic reprogramming (such as expression of the relevant proteins and physiological data showing the effects of reprogramming), co-evolution (e.g. phylogenetic analyses and statistical tests for co-evolution), and effects of SUP05 phages on nitrogen and carbon cycles is needed to provide support for these unnecessary inflationary statements*.

These statements were softened in the revised manuscript. However, we prefer to keep some notion of “virus-host co-evolutionary dynamics” as we believe the environmental genomic analyses of 186 microbial and viral metagenomes using these novel phage genome references offers an unprecedented time series analysis in nature. This revealed the SUP05 dsrC link, where the hosts of these viruses are known (contrast to the Anantharaman paper), and helped document “evolution in action” of one of the phage genomes as well as remarkable conservation of the other (Figure 3) – both of which are new and unique observations for environmental phages, so we hope worth being emphasized in the Abstract.